# How Much Is an Abortion Worth? Was a Human “Not Formed”? An Italian Proposal

**DOI:** 10.3390/healthcare11131948

**Published:** 2023-07-05

**Authors:** Maricla Marrone, Benedetta Pia De Luca, Fortunato Pititto, Ignazio Grattagliano, Nicola Laforgia, Antonella Vimercati, Alessandro Dell’Erba

**Affiliations:** 1Interdisciplinary Department of Medicine (DIM), Section of Legal Medicine, University of Bari “Aldo Moro”, 70124 Bari, Italy; dottfortunatopititto@gmail.com (F.P.); alessandro.dellerba@uniba.it (A.D.); 2Interdisciplinary Department of Medicine (DIM), Section of Psychology, University of Bari “Aldo Moro”, 70124 Bari, Italy; ignazio.grattagliano@uniba.it; 3U.O.C. Neonatology and NICU Policlinico Bari, University of Bari “Aldo Moro”, 70124 Bari, Italy; nicola.laforgia@uniba.it; 4U.O.C. Gynecology and Obstetrics Policlinico Bari, University of Bari “Aldo Moro”, 70124 Bari, Italy; antonella.vimercati@uniba.it

**Keywords:** abortion, indemnity, human capital, fetus, malpractice assessment

## Abstract

Introduction: “Human capital” is defined as an integration of innate skills and knowledge acquired by investing in the formation of an individual; it is a real “capital” that pays off in the long term. In the Italian legal system, a human being is recognised as a “person” from the moment of birth. This determines the acquisition of the personal rights of an individual. Necessarily, therefore, by law, a fetus does not own such rights; nevertheless, it has an innate “potential” to acquire such rights after birth. Objective: In Italian jurisprudence, in general, the damage from a loss of a parental relationship is justified by the condition of existential emptiness caused in the family by the loss of a child. Compensation for this damage in the event of abortion due to third-party responsibility presents a non-uniform recognition in the judgements of the Italian courts, but in any case, it is almost always recognised with limitations since the emotional relationship with the lost individual is defined only in terms of “potential”. Consequently, in this matter, at least two questions can be raised: (i) Is the economic estimate of abortion based on objective and standardised criteria, or is it heavily influenced by subjective evaluation? (ii) Is it possible to find standard criteria that may act as guidelines to quantify the loss of that human capital “in progress”? Methodology: The authors try to answer these questions by analysing the different approaches to this issue adopted at an international level. Conclusions: In conclusion, the authors propose homogeneous criteria to quantify the damage caused by abortion.

## 1. Introduction

Each person has his own capital and, therefore, his own economic value. There is dissimilarity in the valuation of human capital and its economic quantification at the international level. There are also unexplored areas, such as the death of a product of conception. Despite appearing simplistic, the approach adopted in civil forensic medicine to quantify the loss of the “human capital” of an individual is based on the calculation of its “economic value”, which can be determined by referring to standardised tables. This “capital” is based on innate talents and educational and experiential competencies, and, like any other capital, it yields a “product” in the short, medium, and especially long term [1,2].

This concept clearly derives from an economic approach: each individual, based on their innate predisposition, combined with knowledge and competencies acquired during life, generally speaking, can be assumed valuable by an external investor who can decide to invest in this “human capital” to gain profits or results of economic relevance [3].

All of these parameters affect forensic evaluation, the customisation of damages, as well as the premiums of insurance policies and compensation [1,4]. This is well-known in legal–medical doctrine and practice.

The death of an individual resulting from an illegal act determines the damage (with legal relevance) that occurred not only to the direct victim but also to other parties, such as parents and relatives, who can suffer deep distress and deterioration of their daily well-being.

As stated by the Italian Court of Cassation, the loss of a relative “constitutes damage that goes beyond the mere pain that the death of a loved one, especially if preceded by agony, causes in the surviving relatives, consisting in the suffering for being unable to enjoy the presence of the deceased person. This condition results in the irreparable destruction of a life system based on affection, sharing, and reassuring everyday relationships between wife and husband, mother and son, brother and brother, in being unable to do what one has done for years, as well as in the changes that such a loss inevitably produces in the relationships between survivors” (Italian Civil Cassation Court, Section III, Order n. 9196/2018).

In agreement with these statements, Italian case law has developed the concept of “damages for loss of the parent-child relationship”, which is included among the categories of non-monetary damages provided by article 2059 of the Italian Civil Code and compensable to the relatives of a deceased person due to a third-party fault.

Following the aforementioned jurisprudential position, in Italy, indicative criteria have been formulated for the compensation of damages resulting from the loss of the parental relationship, which take into consideration the different family relationships that are theoretically eligible for this purpose. In particular, given that this concerns the violation of rights inherent to the person lacking the possibility of a direct economic conversion, the compensation of damage must consider:The “intensity” of the family bond (the damages will be directly proportional to the degree of kinship);The cohabitation situation;The numerosity of the family unit (the fewer the surviving relatives, the greater the suffered damages);Lifestyle habits;The age of the victim (the damage will be lower the older the victim was at the time of the event) and of the surviving individuals (the damage will be higher the younger the age of the individual).

Regarding the international scenario, although in all nations, there is a distinction between “patrimonial” and “non-patrimonial” damages, the criteria for their compensation vary widely, and there are no universal parameters that can be used for this purpose.

The following tables summarise the quantification methods of damages for death in different countries, i.e., Spain (Table 1), France (Table 2), Germany (Table 3), England (Table 4), and the United States (Table 5). The most significant countries in the assessment of death damages were chosen.

In none of these countries/states, not even in Italy, is there a standardised value of damage from the loss of the product of conception.

In fact, in the Italian legal system, the human being is recognised as a ‘person’ from the moment of birth, i.e., after performing the first respiratory act. In legal terms, this determines the acquisition of rights proper to the person.

Necessarily, therefore, a foetus does not have such rights; yet, on the one hand, he has an innate ‘potential’, and, on the other hand, it is undeniable that on a psychological and biological level, the relationships that parents establish with the product of conception already have value during maternal gestation from its inception and in a progressive manner.

Considering that, according to Italian law, the damage from the loss of a parental relationship is the damage resulting from the loss of a close relative and from the condition of existential emptiness that ensues in the relatives, can the death of a foetus be considered differently from the death of a ‘person’?

Compensation for damages arising in the event of abortion due to the liability of third parties is not uniformly recognised in Italian jurisprudence but is almost always recognised in a partial manner since the emotional relationship with the lost person, recognised by jurisprudence, is defined as only “potential”.

Moreover, if such a bond exists even before birth, the loss of a child should be recognised even before its birth; in fact, the economic recognition of a loss is linked to the injury of a constitutionally guaranteed right. In the present case, the right injured by the loss of a child before its birth is protected by Articles 2 (Article 2 of the Italian Constitution, which came into force on 1 January 1948: “*The Republic recognizes and guarantees the inviolable rights of man, both as an individual and in the social groups where his personality takes place, and requires the fulfilment of the mandatory duties of political, economic and social solidarity*”), 29 (Article 29 of the Italian Constitution, which came into force on 1 January 1948: “*The Republic recognizes the rights of the family as a natural society founded on marriage. Marriage is ordered on the moral and legal equality of the spouses, with the limits established by law to guarantee family unity*”) and 31 (Article 31 of the Italian Constitution, which came into force on 1 January 1948: “*The Republic facilitates with economic measures and other provisions the formation of the family and the fulfilment of related tasks, with particular regard to large families. It protects maternity, childhood and youth, favouring the institutions necessary for this purpose*”) of the Italian Constitution.

In fact, in order to foster the full development of the human person, the Constitutional Charter tends to guarantee the main rights and ethical-social relations, which include family relations and, therefore, also includes the affection that parents develop towards the product of conception even before birth.

Notwithstanding such evidence, there remains no proposed standard of assessment to be inspired by and homologated in order to estimate the “human capital” of the foetus, which therefore suffers exclusively from analogical presumptions that inevitably differ depending on the adjudicating court and the judge himself, as well as on the specific case.

The authors, after having already addressed the issue by providing a qualitative analysis, here attempt to propose quantitative criteria as well in order to create homogeneity in the judge’s assessment.

In a previous study, 31 judgements on compensation for traumatic abortion in Italy were analysed [4]. The analysis revealed a lack of uniformity in the definition of compensation criteria. In spite of this, a common rationale is recognisable in considering the damage as harm from the loss of the parental relationship, albeit potential, created between mother and unborn child. This includes a twofold nature of the damage: one of a “moral” nature, which takes into account the psychological suffering suffered as a result of the bereavement, and one “existential”, understood as dynamic relational damage. Therefore, with a view to guaranteeing uniformity in the assessment of the various Italian Courts, a proposal to define objective parameters that take into account the scientific aspects linked to the loss of the unborn child was generated. Among the criteria identified by the authors are the gestational age at which the loss occurs, any gynecological pathologies and/or other morbid states capable of altering the procreative capacity of the parent, the family bond, the age of the parents, the existence of other children of the couple, and any previous abortions (Table 6). These parameters must recognise the damage suffered in the concrete possibility of establishing a family (a value protected by the Italian Constitution and must compensate for the damage in its moral aspect [4].

The authors will analyse, through the available scientific data, the most relevant and useful items to standardise an evaluation of damage from abortion, thus providing more precise and circumscribed evaluation parameters [4].

We will, therefore, briefly review what has already been undertaken in order to be able to evaluate these parameters more in-depth. These parameters will be analysed individually in order to propose a quantitative approach for each of them.

Given the complexity of the assessment process, medico-legal investigations will have to be based on clinical diagnostic models based on the best scientific evidence in order to minimise the consultants’ margins of approximation and subjectivity. Moreover, an integrated and interdisciplinary approach will be indispensable through a collegial approach involving the collaboration of specialists from different fields (forensic medicine, gynecology, psychology, psychiatry, and child neuropsychiatry).

Having assessed what is necessary to attempt to propose the first quantitative delimitations of the damage caused by the loss of the product of conception, the authors have decided to tackle the issue by going into more detail on the strictly psychological–psychiatric issue (psychological impact of the loss suffered) and on the gynecological issue (maternal age, gestational age, type of pregnancy, the existence of maternal pathologies capable of affecting pregnancy, previous miscarriages, etc., with repercussions on the psychological assessment) [5,6].

We perform this assessment because it is desirable to achieve a shared practice among the various professionals working in this field in virtue of the need to outline rigorous and objective methods of investigation to guide medico-legal work and adjudicators.

The present work has, therefore, the objective of examining the scientific contributions in order to outline a possible methodology of investigation that establishes the prerequisites for the quantification of abortion damages that is correct and as uniform as possible.

## 2. Materials and Methods

### 2.1. Epidemiology of the Loss of the Product of Conception Applied to the Medico-Legal Context

The authors consider it appropriate to state that the quantification of the harm to be addressed in this article refers to the loss of the product of conception during the entire period of pregnancy. In fact, the definition of abortion varies in different legislations. The WHO in 2001 defined spontaneous abortion as the delivery of a foetus that dies below 22 weeks gestational age or is diagnosed as foetal death before 22 weeks without taking into account the time of the expulsion of the foetus and with a neonatal weight of less than 500 g. In Italy, spontaneous abortion is defined as the termination of pregnancy within the 180th complete day of amenorrhoea [7]. In the United Kingdom, this limit is set at 24 weeks and 0 days of gestation. The American Society for Reproductive Medicine defines miscarriage as a clinical loss of pregnancy of less than 20 weeks gestation. The European Society of Human Reproduction and Embryology defines miscarriage as pregnancy loss before 22 weeks gestation [8].

Considering this, for our purposes, the definition that best expresses the concept of loss of the product of conception is ‘death in utero’ as defined by the World Health Organization (WHO). In fact, the WHO definition includes “any death prior to the expulsion or extraction of a product of conception, regardless of the duration of pregnancy” (WHO, ICD, 2004) [9]. According to the WHO, death in utero is confirmed ‘by the absence of respiratory movement or heartbeat of the foetus, or by the perception of the beating of the umbilical cord, or by voluntary muscle contraction, without the prerequisite of gestational age’. Instead, the International Classification of Diseases (ICD) distinguishes early foetal deaths (between 500 and 1000 g or between 22 gestational weeks and 28 gestational weeks) from late foetal deaths in utero (above 1000 g or after 28 gestational weeks) [10].

### 2.2. Psychic Framework

There is no direct and proportional correspondence between the damaging event and the psychic consequences, but the impact that the trauma has had on that specific individual must be considered on a case-by-case basis, according to the meaning he/she has attributed to it and on the basis of his/her aspects of vulnerability and resilience [11].

In fact, the pathogenic charge of a psychotraumatising event cannot be assessed only in relation to the event’s own characteristics (intensity and duration) but must be considered above all in relation to a series of the internal variables of the individual (developmental stage, cognitive capacities, emotional resources, previous traumas, pre-existing psychopathology, etc.).

First and foremost, an accurate anamnestic-circumstantial investigation will be required, which will make it possible to acquire, by means of an interview, the historical and biographical knowledge about the individual, i.e., to carry out an idiographic analysis of the subject’s bio-psycho-social set-up. This interview, where possible, should be supplemented with the examination of clinical documentation and clinically oriented witness statements confirming the existence of a state of psychic alteration.

In this context, ‘relational anamnesis’ is fundamental, i.e., the analysis of the affective covalences that bound the survivor to the deceased, in order to assess whether the anomalous interruption of the relationship gave rise to particular psychodynamic mechanisms that may have destabilised his psycho-behavioural structure [12].

It will then be necessary to assess the psycho-stressful intensity of the bereavement event. In this regard, it should be pointed out that in the case of sudden and/or violent death, as may be the case with an abortion provoked by a third party, there is an increased risk of developing psychopathology, as well as a so-called ‘complicated’ bereavement and, consequently, the structuring of psychic damage.

It should be borne in mind that if the child is aware of the loss of what was to become his or her brother/sister, complicated grief is more likely to be structured within the child.

Complicated grief is characterised by heavy intrusive thoughts concerning the traumatic nature of death, anger, loss of trust in people, flashbacks, nightmares, and terror of imminent catastrophes.

In addition, this is associated with an increased suicide risk and the inability to rebuild a fulfilling emotional life. In the DSM-5, there is a nosographic entity called ‘complicated persistent bereavement disorder’.

The mother who has personally suffered an abortion and ‘witnessed’ the death of the foetus undergoes stressful and traumatic stimuli (e.g., the arrival of the police and medical personnel, ‘premonitory’ symptoms of the subsequent abortion, etc.), which, when encoded as intrusive memories, interfere with the possibility of activating those resources necessary to deal with the grieving process. Often, moreover, the parent may feel a sense of guilt because he or she was incapable of doing anything.

The next step will be to operate a nosographic framing that allows the symptomatology expressed by the bereaved to be classified within defined psychopathological frameworks, enabling a diagnosis to be made. In spite of the fact that exposure to a traumatic event produces heterogeneous reactions, according to consolidated clinical and medico-legal experience, depressive, anxiety, bipolar, and traumatic-stress-related disorders are mostly found following a bereavement that is not adequately processed [13,14,15].

In minors, on the other hand, it is relatively rare to detect those disorders typically described for adults, but more frequently, in pre-school and latency age, regression symptoms (functional disturbances in sphincter control, speech, and school learning, i.e., somatoform symptoms through which the child more or less consciously seeks greater attention) can be found. In preadolescence and adolescence, behavioural reactions of withdrawal, isolation, aggression, and self and hetero-directed anger are reported [16].

To complete the classic investigation (anamnesis, clinical interview, and observation), an accurate and specialised psychodiagnostic examination appears to be necessary, which makes use of tests that prove capable of obtaining valid, replicable, and stable results over time.

Lastly, it is necessary to point out the importance of gathering the information needed to frame the assessed subject’s functioning from the people around him/her. This will be especially necessary when the subject to be assessed is a minor. In this case, in fact, the evaluator will have to listen not only to the child but also to family members, teachers, and specialists who have observed the child before him/her in order to outline a picture of the subject’s functioning in their various living environments.

### 2.3. Gynecological Assessment

It is important to know the probability of the miscarriage of a specific pregnancy on a case-by-case basis in order to guide the assessor as to the mother’s chances of carrying that pregnancy to term. It is equally important to be able to know the risk of maternal miscarriage (i.e., for any pregnancy) in order to understand the possibility of enjoying future pregnancies, which would make it possible to understand the ‘preciousness’ of a pregnancy terminated due to a third party. In this sense, it is equally important to give an indication of the difficulty of past and future conception based on pathologies that concretely hinder the possibility of having a child.

Quantitatively assessing parameters, such as maternal age, previous miscarriages, the type of pregnancy, and the existence of maternal morbid pathologies affecting the development of the pregnancy, will make it possible to better calibrate the relative economic assessment of the loss of the product of conception.

From a methodological point of view, there are objective difficulties in designing accurate studies that define the abortion rate with certainty. Indeed, many abortions occur before the pregnancy is even known to the mother. Nevertheless, the data obtained by monitoring women actively seeking pregnancy do not reflect the actual population. Therefore, it is difficult to define a percentage value of the phenomenon that is unambiguously representative of reality [17].

Although the cause of most miscarriages is unknown, it is likely that most miscarriages result from the interaction of age, genetic, hormonal, immunological, and environmental factors. Among these, maternal age plays an important role [18,19]. Data from the Norwegian national registers (birth register, induced abortion register and patient register) report a biphasic relationship between maternal age and abortion, with a J-shaped pattern, shown in Figure 1.

In fact, the abortion rate is slightly higher in pregnancies started before the age of 20 and increases considerably above the age of 35. In particular, it emerges that the risk of miscarriage is lowest among women aged 25–29 (9.8%), with the lowest risk at age 27 (9.5%) and the highest at age 45 and over (53.6%). Table 7 shows the complete study data [20].

The likelihood of miscarriage also varies in relation to gestational age. A systematic review by Avalos et al., 2012 analysed data from the first 20 weeks of pregnancy, finding higher abortion rates before week 12, which then decreased after week 14. As already mentioned, data collection is very much influenced by the methodology of the study, i.e., the mode of the reference sample, as well as the mode of diagnosis of pregnancy and abortion. This explains the wide variation in results between weeks 5 and 13 depending on the authors. The full data are shown in Table 8 [17].

There are fewer studies analysing the phenomenon after the 20th week of gestation. To the best of our knowledge, there are no data indicating the percentage of fetal mortality in relation to the specific gestational week. Aggregate data collected by Quibel et al., 2014 [10] on death in the third trimester estimate a prevalence of intrauterine foetal death at 2% worldwide, with consistent data for high-income countries around 4–5/1000 births and much more variable data for low-income countries [10].

It is known that in the presence of a history of previous miscarriage, the risk of not carrying the current pregnancy to term increases gradually (odds ratio of 1.5 after a first miscarriage, 2.2 after two miscarriages, and 4.0 after three miscarriages) [20]. In addition, there is a moderate increase in the risk of abortion if the previous delivery was by Caesarean section (odds ratio of 1.16, 1.12–1.21) [20].

Maternal health status can also affect the outcome of a pregnancy, and several risk factors have been identified that can increase the risk of miscarriage. These can operate either directly by exerting their effects on the foetus or indirectly, i.e., by aggravating maternal health status and adversely affecting the continuation of the pregnancy.

Among these risk factors, high blood pressure (chronic or acquired during pregnancy) can lead to vascular complications, both for the mother (e.g., pre-eclampsia) and the foetus (e.g., intrauterine growth retardation). A meta-analysis by Flenady et al. assessed the impact of chronic hypertension on the risk of foetal death, which is more than doubled in the presence of this condition (2.6, 95% CI 2.1–3.1). Furthermore, conditions acquired during pregnancy are also associated with an increased risk; in particular, pregnancy-induced hypertension has a 1.3-fold increased risk (95% CI 1.1–1.6), pre-eclampsia has a 1.6-fold increased risk (95% CI 1.1–2.2), and eclampsia has a 2.2-fold increased risk (95% CI 1.5–3.2) [10]. In contrast, there was no significant association between a history of pre-eclampsia in the previous pregnancy and the risk of miscarriage [20].

A moderately increased risk of miscarriage was found in relation to gestational diabetes in a previous pregnancy (odds ratio of 1.19, 1.05–1.36) [20].

The risk of miscarriage also increases in the presence of an obese condition (defined as a BMI ≥ 30 kg/m^2^). In fact, obesity is an independent risk factor for recurrent abortion (OR 1.75, 95% CI 1.24–2.47) [21].

Polycystic ovary syndrome (PCOS) is not in itself a direct risk factor for abortion. However, it is often associated with comorbidities (obesity, metabolic syndrome, hyperinsulinaemia, or hyperandrogenemia) that increase the risk of abortion. The prevalence of PCOS in women with recurrent abortions is similar to that of the general population, and PCOS status does not influence the prognosis of recurrent abortion [21]

Thrombophilia is a family of acquired or inherited disorders that involve alterations in blood clotting processes, with an increased risk of developing venous or arterial thromboembolism. Among hereditary thrombophilias, the most frequent are associated with factor V Leiden or protein C, protein S, and anti-thrombin deficiency. Although early studies showed an association between recurrent miscarriages and hereditary thrombophilia, more robust analyses have not confirmed these associations [21]. Instead, among acquired thrombophilias, special mention must be made of antiphospholipid syndrome (APS), which is the most common treatable cause of recurrent abortion (15–20% of cases). APS is characterised by vascular thrombosis and/or obstetric complications, including early and late abortions (<20–24 weeks gestation) and third-trimester complications such as pre-eclampsia, preterm delivery, and fetal growth restriction. Miscarriage is the most common complication of APS in pregnancy, occurring in 38.6% of women with APS [21].

Rheumatoid arthritis (RA) is a chronic systemic autoimmune disease. A 2022 meta-analysis, which analysed 18 studies involving over 50 million cases, reported a moderately increased risk of miscarriage (OR, 1.16; 95% CI 1.04–1.29) [22].

The role of male age on the probability of conception is less studied and less clear than that of female age [23]. Older male age has been associated with a reduction in sperm volume, total sperm count, sperm motility, and normal sperm morphology. In addition, there was a gradual change in testicular vascularisation and a regular decrease in the number of Sertoli and Leydig cells, with an associated decrease in serum testosterone [24]. A recent systematic review of the literature showed that advanced paternal age is associated with subfecundity and a longer time to pregnancy [23]. In a 2020 meta-analysis, increasing paternal age was shown to be correlated with an increased risk of miscarriage. Specifically, assuming a group of men aged 25–29 years, the risks were 1.04, 1.15, 1.23, and 1.43 for the age groups 30–34, 35–39, 40–44, and ≥45 years, respectively [25].

The role of paternal age is also implicated in the probability of conceiving offspring affected by genetic disorders. Indeed, many genetic disorders in offspring are correlated with conception at an advanced paternal age. The DNA quality of the sperm of older men may be affected by germ cell mutations that accumulate over the years, as well as by the impairment of DNA repair mechanisms [26]. The incidence of several autosomal dominant diseases such as achondroplasia, colonic polyposis, Marfan syndrome, Apert syndrome, or basal cell nevus are associated with advanced paternal age, whereas there is no clear evidence of a paternal effect on structural or numerical chromosomal abnormalities, with the exception of trisomy 21 [27]. In fact, even if there is a slightly increased incidence of certain genetic diseases among children born to older men, the individual risk of such a new disease is still very low since these are low-incidence diseases [26].

Finally, when defining the likelihood of carrying a pregnancy to term, account must be taken of the fact that, as shown by Wang et al., an increasing number of pregnancies are achieved by means of artificial fertilisation methods (medically assisted procreation, or PMA) [28]. PMA success rates are affected by several factors. As with physiological pregnancy, maternal age is the most important factor associated with increased rates of miscarriage. The statistical association between maternal age and abortion rate in a pregnancy achieved by PMA techniques becomes significant from the age of 32 years (OR 1.2; 95% CI 1.2–1.2; *p* < 0.001). Below 32 years of age, however, this association is not statistically significant (OR 1.0; 95% CI 1.0–1.0; *p* = 0.060) [29]. Compared with women under 30 years of age, the risk of abortion between 36 and 40 years is increased (HR = 1.63, 95% CI = 1.28–2.08, *p* < 0.001). The risk is even higher for women over 40 (HR = 4.14, 95% CI = 2.63–6.52, *p* < 0.001). In general, the risk gradually increases as maternal age increases, reaching an abortion rate of 37.7% above the age of 40 [30]. In the case of PMA, male age may also play an important role and may be characterised as an independent risk factor [31].

The probability of abortion is increased 1.1-fold for every year of increase in male age [29].

In the case of PMA, maternal pathologies may also play a role. For example, the association between female obesity and the risk of abortion after 12 weeks of gestational age has been shown (HR = 1.52, 95% CI = 1.11–2.10, *p* = 0.010) [30].

## 3. Results and Discussion

The death of a person as a result of an unlawful act determines the damages (of legal significance) accrued not only by the person who is the direct victim but also by other persons, relatives of the injured party, who suffer profound suffering and alteration of their daily routines.

In Italy, damage from the loss of parental relations (non-asset damage) is provided for in Article 2059 of the Civil Code and is compensable to the relatives of a deceased person due to the wrongful acts of others.

However, this applies to ‘born’ individuals. Instead, for the unborn, no specific assessment has yet been proposed: in fact, there are no standard references on which to calculate the value of the ‘human capital’ lost.

When a judge has to decide how much to compensate in the event of the loss of the product of conception constituting a crime, in most cases, there is a tendency to reduce the economic estimate proposed by the already well-known “*barhemes*” in consideration of a relationship in ‘potency’ rather than in ‘act’.

The difference is probably rooted in the wording of Article 1 of the *Civil Code*, paragraph II, which states that the rights of the conceived recognised by law are subordinate to the event of birth (and this notwithstanding the openness to the recognition of the ‘rights of the unborn child’ contemplated in certain exceptional regulations).

It is in these terms, therefore, that human capital should be considered with specific parameters even for what is ‘innate’, that is, what exists regardless of the unfolding of later life.

Therefore, according to the prevailing case law, although the prejudice (from abortion) should be recognised as the loss of the “potential” parental relationship with the unborn child, when translating this “damage” into economic terms, it should be calibrated on the basis of specific factors that allow it to be personalised so that it can, in certain circumstances, even come close to the recognition of damage from full parental loss.

### Proposal of Parameters for Assessing Damages for Parental Loss in Case of Loss of the Fetus in the Italian Context

The Italian Supreme Court of Cassation, in its decision no. 33005 of 10 November 2021, affirmed that the injury deriving from the loss of a close relative could not be compensated on the basis of the “Milan Court tables” in force at the time, which did not meet the requirements indicated by case law (civil cassation 10579/2021; civil cassation 26300/2021). The Milan tables, in fact, in the liquidation of the damages in question, did not at the time follow the “point technique” but identified a minimum ground and a maximum ceiling, between which there is a significant difference in the absence of any indication of determined criteria for establishing what amount to liquidate. In fact, in the definition of the quantum compensable, it is necessary that the liquidation be based on the variable point technique and take into consideration indefectible circumstances such as the age of the victim, the age of the survivor, and the degree of kinship and cohabitation exactly as provided by the Roman tables. As a result, on 28 June 2022, the *General Criteria for the Liquidation of Non-asset Damage Deriving from the Loss of Parental Relations—Integrated Tables with Points—Edition 2022* were formulated by the Civil Justice Observatory of Milan: in particular, from the monetary values provided by the Milanese tables, a “point value” system was derived for the case of the loss of parents/children/spouse/assimilates, as well as for the case of the loss of siblings/grandchildren, respectively, of EUR 3365.00 and EUR 1461.20. A system for the distribution of points was then proposed according to the factual parameters indicated by the Court of Cassation (the age of the primary victim and the secondary victim, cohabitation between the two, the survival of other relatives, and the quality and intensity of the specific affective relationship lost).

The Italian Court of Cassation, therefore, with two new sentences (33645/2022; 37009/2022), has “promoted” such new tabular criteria specifying that the criteria have a subjective nature and concern both the dynamic relational aspects (upheaval in the life of the secondary victim as a consequence of the loss) and those of internal suffering, both of which must be attached and proved, also by presumptions, since it is not possible to predict, in the civil liability system, the existence of a case of damage *in re ipsa*.

On the basis of these new indications, an evaluation proposal must inevitably be formulated that extends to the “third party liability” loss of the foetus. In particular, the above table will be recalled in its entirety, but it will be readjusted to the parameters which, according to the authors, will be necessary to assess in order to quantify the loss of the product of conception. These parameters will also be broken down on the basis of evidence from the scientific literature.

However much one may try to standardise certain evaluations, they will always be affected by a subjective criterion, as they cannot always fully represent the real damage suffered.

The limitation of the study remains the impossibility of understanding every possible evaluative element related to the individual case.

Taking into account what is reported in the previous tables, it is now necessary to analyse and explain the reasons, which are not immediately deducible, behind some of the criteria.

In particular, the association of the scores to the trimesters of pregnancy, present in criterion (A) of both tables, was carried out according to the abortion data present in the systematic review by Avalos et al. (2012) [17], explained above. In contrast, the scores assigned according to maternal age are correlated with the abortion rates related to the woman’s age and extensively discussed in the study by Magnus et al. (2019) [20], also reported previously. Those associated with paternal age are related to the reduction in paternal fertility, which declines progressively with age by virtue of a reduction in sperm quality and motility [23]. Finally, the association of the scores with the age of the brother/sister is made, taking into account the life span they could have shared with the future brother/sister.

With regard to the evaluation of psychic suffering, included in criterion (E) of both tables, it should be noted that it is not a question of psychiatric damage documented by specialists (in fact, it will eventually be evaluated as damage in its own right and with a percentage of biological damage), but rather of demonstrating the existence of suffering of a psychic type peculiar to the specific event/offence, which may be considered as a plus in the evaluation of the damage from the loss of the product of conception.

Here it is that, on the basis of the analysis of the elements most frequently used in the liquidation estimates formulated in the sentences (cited above) recognising the damage from parental loss in the case of loss of the product of conception, albeit on the basis of logical-scientific evaluations, one could propose the equivalent of the calculation of the damage from parental loss drawn up by the Court of Milan (for the “born” subject) for the foetus.

In the points system, the economic consideration of the damages is determined by assigning a numerical score to be multiplied by a sum of money that constitutes the ideal value of the single point of non-asset damage.

The point system should therefore take into account elements specifically related to the degree of “suffering” conceivable for the loss of the foetus, as illustrated in Table 9 and Table 10.

## 4. Conclusions

The absence of standard criteria to favour an assessment closer to objectivity in cases of parental loss resulting from the loss of the product of conception due to the liability of third parties is an obvious international shortcoming.

For this reason, it would be necessary to associate national guidelines drawn up by consensus conferences, constituted by scientific societies with national diffusion (Italian Society of Legal Medicine and Insurance, Italian Society of Psychiatry, Italian Society of Criminology, Italian Society of Child Neuropsychiatry), to standardise the criteria and clinical and psychodiagnostic protocols for evaluating bereavement damage in cases of parental loss deriving from abortion on the responsibility of third parties.

This inevitably gives rise to differences in the national territory, with differences linked to the subjective assessment of the judges, who do not in all cases consider all the circumstantial/familial elements in which the harmful event is determined, and the omission of certain parameters generates an inevitable compensation prejudice.

The proposal of a homologated criterion, or rather of evaluative elements on which to base the liquidation of damages, is therefore functional for the purpose of remedying existing discrepancies, thus providing an objective and parameterised aid to the judge, but also to the medico-legal expert.

Moreover, it will leave—as already happens in any area of liquidation of non-asset damage—a space for the subjectivity of the evaluator, but it will orient the final estimate on fixed points, increasing its predictability.

If this proposal may be useful in the Italian context, it may also be useful in other jurisdictions to assess the possibility of assigning a ‘standardised’ economic value to the loss of the product of conception.

Allowing judges to have technical elements for compensatory assessments is fundamental in any field. It is even more important when it comes to compensating for the loss of a loved one, even if not yet born. Trying to prevent such assessments from being a consequence of arbitrariness will allow judges to emphasise ‘justice’.

If human capital is defined as an integration of innate endowments and acquired knowledge, the foetus is also the bearer of a capital that cannot be eluded but rather is appreciable and therefore compensable.

## Figures and Tables

**Figure 1 healthcare-11-01948-f001:**
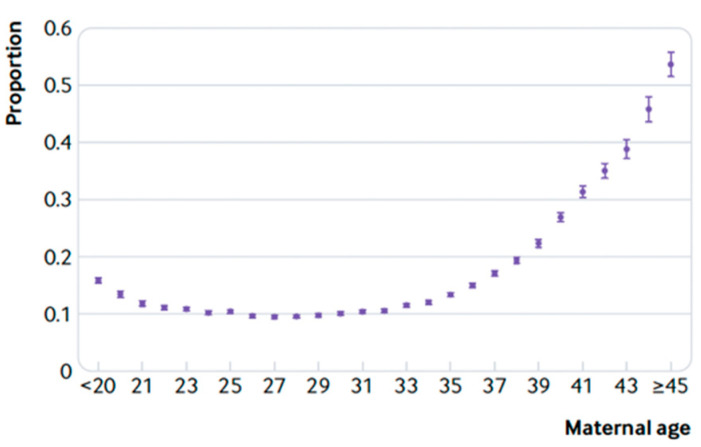
Risk of miscarriage according to maternal age. The bars for each point reflect the 95% confidence intervals (table extracted from Magnus et al. [20]).

**Table 1 healthcare-11-01948-t001:** Spanish scenario.

Spain
In Spain, Tabla I of the Baremo regulates the criteria for compensation, including the quantification of non-material damages, basic pecuniary damages, and the legal determination of the injured parties; it also establishes the criteria for exclusion and concurrence. In quantifying the damages to be paid to the heirs, the relationship of kinship with the victim and the victim’s age are considered. These values are then corrected by Table II, which establishes percentages of increase or decrease depending on the victim’s income. The values in the different Tablas are updated each year to take account of inflation. Compensation in Spain is, therefore, homogeneous and linked to income. Finally, medical care and funeral expenses are compensated separately, in addition to the Baremo.

**Table 2 healthcare-11-01948-t002:** French scenario.

France
In France, the “non-patrimonial” damages to be recognised in the event of death are addressed in the Dintilhac Report of 2006 (updated in 2013 with the “*Indemnisation des dommages corporels. Recueil méthodologique commun*”). In particular, two types of damages are recognised: (a) “*préjudice d’affection*”, which is the suffering incurred from the death of a relative, and (b) “*préjudice d’accompagnement*”, which is the damage derived from the need for assistance provided to a person from the moment of the accident or onset of illness until death. “*Préjudice d’affection*” is recognised for parents, children, grandparents, grandchildren, spouses, and cohabitants. Recognition of this damage for people other than these can only be granted if the emotional bond with the victim is demonstrated. No compensation is provided for the deceased, except for the equivalent of the Italian “damage from lucid agony”, which is the suffering experienced in recognising the imminence of one’s death, which a recent French Supreme Court ruling recognised as a separate item of damages from what is provided for in the Dintilhac Report’s nomenclature (Cour de Cassation, Chambre Mixte, 25 mars 2022, n° 20-15.624).

**Table 3 healthcare-11-01948-t003:** German scenario.

Germany
Non-patrimonial damage is regulated by Article 253—Immaterieller Schaden of the German Civil Code (BGB), which admits that “*in the case of damage that is not pecuniary damage, pecuniary compensation may be claimed only in the cases provided for by law*”; the second paragraph then states that damage to the body, health, freedom, and sexual self-determination may also be compensated. However, there are no tabular criteria for determining the quantum; the assessment and quantification of damages are made by the judge, who makes use of case law precedents, which, however, are not binding. This assessment is influenced by the severity of the injuries, the age of the victim, the intensity of the pain, and other factors that may play a role in the severity of the damage. That said, it should be noted that in Germany, the recognised type of ‘non-asset’ damage is the ‘*Schockschaden*’. Compensation is due only to the close relatives of the deceased (parents, spouses, children), who, in addition to the emotional damage linked to the loss of the relative, have suffered psychological damage that, transferred to the Italian system, is assimilated to real biological damage, linked to psychiatric pathology (e.g., depression or post-traumatic stress disorder) deriving from having witnessed the death of the relative. However, the legislator’s intervention has evolved this concept, and since 2017, Article 844 paragraph 3 of the BGB has come into force, which, in the event of homicide, grants the close relatives of the deceased (spouse, cohabiting partner, parent or child) compensation in the form of a survivor’s indemnity for ‘psychic suffering’, which, in turn, can be classified as suffering from bereavement, even in the absence of a nosographic pathology.

**Table 4 healthcare-11-01948-t004:** Anglo-Saxon scenario.

England
In England, the 16th edition of “*The Judicial Studies Board’s Guidelines for the Assessment of General Damages*” includes both biological damage itself and damage for “pain and suffering”. In this type of compensation, “non-pecuniary” damage is fully integrated.

**Table 5 healthcare-11-01948-t005:** American scenario.

United States
In the United States, on the subject of non-economic damages, compensation for “loss of society and comfort” is denied by many statutes, but where it is recognised, it leads to higher damages. The items of damages are variable and also include love and affection. Some jurisdictions allow “loss of guidance and advice” when a parent dies (thus leaving the children without any guidance). However, most cases deal with economic losses, and, in most jurisdictions, the survivors are paid what they would have received if their relative had not died. For this assessment, life expectancy, future economic income, habits, state of health, the possibility of future salary increases, and possible future depreciation due to inflation are evaluated. Given the objective difficulties in evaluation, juries have a wide margin of discretion in any case. Finally, typical of the US system is the aspect of punitive damages (punitive, exemplary, or vindictive damages), where malice, willful misconduct, or particularly serious offences are found. Finally, just as in the case of the death of a ‘person’, suffering must also be considered on the part of grandparents and/or uncles, who are also involved (to varying degrees according to defined parameters) in the “injury” to the family unit.

**Table 6 healthcare-11-01948-t006:** Parameters for the evaluation of damage from parental loss in the case of abortion.

Parameters for Assessing Damages from Parental Loss in Case of Abortion	
1. Gestational age at which the loss occurs	The earlier the loss, the less damage will be caused.
2. Any gynecological pathologies and/or other illnesses that could alter the parent’s procreative capacity	The presence of other gynecological pathologies and/or other illnesses that could alter the parent’s procreative capacity increases the extent of the damage.
3. Parental relationship	The mother may be entitled to greater compensation than the father, with diminishing amounts for other degrees of relationships, limited to possible siblings, grandparents, and aunts and uncles of the fetus, while also considering the condition of cohabitation/association and thus participation in the loss. For underage siblings, an age constraint could be introduced, taking into account the degree of understanding of the grievous event itself. Additionally, based on the age of the parents, the risk for the eventual “only child” status of the couple’s sibling could be evaluated.
4. Parental age	The older the parents, the greater the resulting damage, as this also considers the reduced chances of further conception.
5. Existence of other children of the couple	The existence of other children would reduce the extent of the damage.
6. Possible previous abortions	The existence of other abortions would increase the extent of the damage, reducing the hypothetical chances of further conception.

**Table 7 healthcare-11-01948-t007:** Table adaptation from Magnus et al. (2019) [20].

Age of Mother	Abortion Rate
<20	15.8%
20–24	11.3%
25–29	9.8%
30–34	10.8%
35–39	16.7%
40–44	32.2%
>45	53.6%

**Table 8 healthcare-11-01948-t008:** Abortion rate as a function of gestational age. Table extrapolated from Ammon et al., 2012 [17].

Weekly Miscarriage Rate (per 1000 Woman-Weeks) and 95% CI by Gestational Age and Study								
Gestational Week	Taylor, 1970		Harlap et al., 1980		Goldhaber and Fireman, 1991		Li et al., 2002	
	Rate	95% CI	Rate	95% CI	Rate	95% CI	Rate	95% CI
5	29.63	(3.59–107.03)	36.70	(4.44–132.56)	4.41	(0.11–24.54)	32.79	(10.65–76.51)
6	5.78	(0.7–20.88)	5.73	(0.14–31.93)	5.48	(2.01–11.93)	21.05	(9.63–39.96)
7	9.02	4.32–16.58)	13.83	(5.07–30.09)	7.03	(4.02–11.42)	28.59	(17.22–44.65)
8	7.14	(4.42–10.92)	6.46	(2.37–14.06)	11.06	(7.66–15.45)	34.62	(22.81–50.36)
9	6.22	4.38–8.57)	11.70	(7.24–17.88)	12.48	(9.07–16.76)	28.55	(18.3–42.49)
10	7.62	(5.97–9.58)	14.51	(10.59–19.42)	15.9	(12.19–20.38)	21.88	(13.17–34.16)
11	11.99	(10.21–14)	16.09	(12.61–20.23)	14.11	(10.77–18.17)	23.96	(14.83–36.62)
12	10.22	(8.76–11.86)	10.59	(8.1–13.6)	14.88	(11.56–18.87)	21.76	(13.1–33.99)
13	6.80	(5.71–8.04)	7.04	(5.19–9.34)	8.58	(6.18–11.6)	11.55	(5.54–21.25)
14	4.38	(3.56–5.33)	3.99	(2.71–5.66)	5.21	(3.43–7.57)	8.11	(3.28–16.8)
15	3.97	(3.23–4.84)	3.49	(2.35–4.98)	4.81	(3.14–7.05)	7.05	(2.59–15.34)
16	2.88	(2.27–3.61)	3.76	(2.62–5.22)	4.31	(2.76–6.42)	1.18	(0.03–6.57)
17	2.66	(2.08–3.35)	1.61	(0.92–2.61)	2.11	(1.09–3.68)	1.18	(0.03–6.58)
18	2.23	(1.71–2.86)	3.61	(2.56–4.96)	1.9	(0.95–3.4)	1.18	(0.03–6.6)
19	2.43	(1.89–3.08)	2.09	(1.32–3.13)	2.73	(1.56–4.43)	2.38	(0.29–8.59)
20	2.15	(1.65–2.76)	2.88	(1.99–4.05)	1.68	(0.81–3.09)	1.2	(0.03–6.71)

CI, confidence interval.

**Table 9 healthcare-11-01948-t009:** Proposal for an integrated points table for the liquidation of non-pecuniary damage from loss of the product of conception for parents and siblings.

Integrated Points Table for the Compensation of Non-Pecuniary Damage Due to the Loss of the Product of Conception for Parents and Siblings	
“Point value” of the integrated points Table 2022: EUR 3365.00	
Distribution of points	
A. **Gestational Age**: up to 28 points for presumed non-pecuniary damage (emotional and dynamic relational distress)	- 1st trimester 🡢 4 points - 2nd trimester 🡢 16 points - 3rd trimester 🡢 28 points
B. **Age of the Secondary Victim**: up to 28 points for presumed non-pecuniary damage (emotional and dynamic-relational distress)	
a. **Mother**	
	- <20 years 🡢 20 points - From 20 to 34 years 🡢 28 points - From 35 to 39 years 🡢 20 points - From 40 to 44 years 🡢 16 points - >45 years 🡢 4 points
b. **Father**	
	- From 0 to 20 years 🡢 2 points - From 21 to 30 years 🡢 4 points - From 31 to 40 years 🡢 8 points - From 41 to 50 years 🡢 10 points - From 51 to 60 years 🡢 12 points - From 61 to 70 years 🡢 14 points - From 71 to 80 years 🡢 16 points - From 81 to 90 years 🡢 18 points - From 91 to 100 years 🡢 20 points
c. **Brother/Sister**	
	- From 0 to 5 years 🡢 20 points * - From 6 to 20 years 🡢 20 points - From 21 to 30 years 🡢 4 points - From 31 to 40 years 🡢 8 points - From 41 to 50 years 🡢 10 points - From 51 to 60 years 🡢 12 points - From 61 to 70 years 🡢 14 points - From 71 to 80 years 🡢 16 points - From 81 to 90 years 🡢 18 points - From 91 to 100 years 🡢 20 points * Given the young age, such points can only be awarded if the child is considered capable of understanding the gravity of the situation and based on their participation in the pregnancy.
C. **Cohabitation**: a. 16 points for presumed non-pecuniary damage (emotional and dynamic-relational distress) in the case of cohabitation between the secondary victim and the product of conception/mother. b. 8 points for presumed non-pecuniary damage (emotional and dynamic-relational distress) may be attributed in the case where the secondary victim and the product of conception/mother live in the same building or condominium complex.	
D. **Survival of Other Primary Family Members of the Product of Conception**: up to 16 points for presumed non-pecuniary damage (emotional and dynamic-relational distress)	- No survivors 🡢 16 points - 1 survivor 🡢 14 points - 2 survivors 🡢 12 points - 3 survivors 🡢 9 points
E. **Quality and Intensity of the Affectional Relationship That Characterised the Specific Parental Relationship Lost**: up to 30 points The quality and intensity of the affectional relationship that characterised the specific parental relationship lost will be taken into account, both in terms of the emotional distress suffered (which may also be presumed) and in terms of the upheaval in the secondary victim’s life (dynamic relational dimension), assessing whether to proceed with the liquidation of parameter D with a single monetary amount or with separate amounts for each of the mentioned items/components of non-pecuniary damage. a. **Frequency/contacts with the mother of the conception product**: absent/sporadic/frequent/always b. **Sharing of holidays/occasions/vacations**: absent/sporadic/frequent/always c. **Sharing of work activities/hobbies/sports**: absent/sporadic/frequent/daily d. **Assessment of psychological suffering**: especially in the case of siblings, it will be necessary to assess their potential involvement in the pregnancy. This point is different from psychological damage understood as biological harm. e. **Assisted Reproductive Technology (ART)**: if the conception product was obtained through ART techniques, it will be a reason for increased damage, as it is possible to assume greater emotional involvement of the parents and greater difficulty in obtaining a future pregnancy. f. **Possibility of having a subsequent pregnancy in relation to the age of the parents**: it should be considered that, for women, as age increases, there will be a lower possibility of having a subsequent pregnancy; therefore, the judge may eventually recognise a plus inversely proportional to the age of the woman. g. **Possibility of having a subsequent pregnancy in relation to the health status of the parents**: if the parents have a pathology of the gynecological system and/or other morbid conditions that can alter procreative ability, it will be necessary to estimate greater damage due to the lower probability of obtaining a future pregnancy.	

**Table 10 healthcare-11-01948-t010:** Proposal for an integrated points table for the liquidation of non-pecuniary damage from loss of the product of conception for relatives in the second degree.

Integrated Point-Based Table for the Liquidation of Non-Patrimonial Damages Resulting from the Loss of the Product of Conception for Second-Degree Relatives	
“Point value” of the integrated points Table 2022: EUR 1461.20	
Distribution of points	
A. **Gestational Age**: up to 28 points for presumed non-pecuniary damage (emotional and dynamic-relational distress)	- 1st trimester 🡢 4 points - 2nd trimester 🡢 16 points - 3rd trimester 🡢 28 points
B. **Age of the Secondary Victim**: up to 28 points for presumed non-pecuniary damage (emotional and dynamic-relational distress)	- From 0 to 20 years 🡢 20 points - From 21 to 30 years 🡢 18 points - From 31 to 40 years 🡢 16 points - From 41 to 50 years 🡢 14 points - From 51 to 60 years 🡢 12 points - From 61 to 70 years 🡢 10 points - From 71 to 80 years 🡢 8 points - From 81 to 90 years 🡢 4 points - From 91 to 100 years 🡢 2 points
C. **Cohabitation**: a. 20 points for presumed non-pecuniary damage (inner suffering and dynamic-relational) b. 8 points for presumed non-pecuniary damage (inner suffering and dynamic-relational) may be awarded in case the secondary victim and the conception product/mother live in the same building or condominium complex c. 25 points for presumed non-pecuniary damage (inner suffering and dynamic-relational) in case of cohabitation for over 30 years between the secondary victim and the conception product/mother d. 30 points for presumed non-pecuniary damage (inner suffering and dynamic-relational) in case of cohabitation for over 40 years between the secondary victim and the conception product/mother	
D. **Survival of Other Primary Family Members of the Product of Conception**: up to 16 points for presumed non-pecuniary damage (emotional and dynamic-relational distress)	- No survivors 🡢 16 points - 1 survivor 🡢 14 points - 2 survivors 🡢 12 points - 3 survivors 🡢 9 points
E. **Quality and Intensity of the Affectional Relationship That Characterised the Specific Parental Relationship Lost**: up to 30 points The quality and intensity of the affectional relationship that characterised the specific parental relationship lost will be taken into account, both in terms of the emotional distress suffered (which may also be presumed) and in terms of the upheaval in the secondary victim’s life (dynamic-relational dimension), assessing whether to proceed with the liquidation of parameter D with a single monetary amount or with separate amounts for each of the mentioned items/components of non-pecuniary damage. a. **Contact/frequency of contact with the mother of the conception product**: absent/sporadic/frequent/always b. **Sharing of holidays/celebrations/vacations**: absent/sporadic/frequent/always c. **Sharing of work activities/hobbies/sports**: absent/sporadic/frequent/daily d. **Assessment of psychological suffering**: it will be necessary to consider an additional amount of damage if there has also been psychological damage. This point is different from psychological damage intended as biological damage.	

## Data Availability

Not applicable.

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
