# Peer review of "How Much Is an Abortion Worth? Was a Human “Not Formed”? An Italian Proposal"

_healthcare, 2023, doi:10.3390/healthcare11131948_

Round 1

Reviewer 1 Report

Dear Authors,

Thank you for this interesting proposal. I am pasting some of my concerns for your action.

  • Insert a citation on the statement in line 37-39 for the subsequent statement to begin with the paragraph's central argument i.e. ‘The human capital'.

  • Delete ‘e.t.c.’ on line 275.

  • On line 300, Delete the word ‘in fact’

  • On Line 318....It is known by who? Kindly revise this statement for clarity.
  • On line 330, revise the statement for clarity. What was/were the key findings from the study by Flenady et al., that either resonate or differ with your study proposal? Kindly include that.

  • On lines 384-385 …..'An increasing number of pregnancies are achieved by means of artificial fertilisation methods'…….contextualize the statements.

  • In the conclusions section, the take-home message should be better emphasized.
  • Ensure that all cited authors are in the bibliography.

 Kind Regards and looking forward to reading this article in the future.

English is fair and to the point

Author Response

Dear Auditor, 
Thank you for your advice. We have amended the text according to your opinion.

  • Insert a citation on the statement in line 37-39 for the subsequent statement to begin with the paragraph's central argument i.e. ‘The human capital'.

We have insert a citation on the statement in line 37-39 for the subsequent statement. 

  • Delete ‘e.t.c.’ on line 275.

We have delete "e.t.c." on line 275.

  • On line 300, Delete the word ‘in fact’

We have delete "in fact" on line 300.

  • On Line 318....It is known by who? Kindly revise this statement for clarity.

We have revise the statement on line 318 for clarity. 

  • On line 330, revise the statement for clarity. What was/were the key findings from the study by Flenady et al., that either resonate or differ with your study proposal? Kindly include that.

We have revise the statement on line 330 for clarity. 

  • On lines 384-385 …..'An increasing number of pregnancies are achieved by means of artificial fertilisation methods'…….contextualize the statements.

We have contextualize the statements on lines 384-385. 

  • In the conclusions section, the take-home message should be better emphasized.

We have better emphasized the take-home message in the conclusions section. 

Kind regards, 

Dr. De Luca Benedetta Pia

Reviewer 2 Report

Dear Authors 

The article " How much is an abortion worth? Was human two capital "not formed"? An Italian proposal "is supported by interesting objectives with interest in legal medicine. The proposal for parameters to assess compensation for parental loss in the event of fetal loss in the Italian context may also be used in other countries with due adjustments and validation.

The study's methodology is adequate. However, the information relating to the methodology chapter must be reorganized since the authors do not present the study results.

Some details might be improved to increase the manuscript's clarity suggested below.

Abstract

Although it is a review article, the abstract should be organized in a structure: Introduction, objective, methodology and conclusions.

Introduction

Line 149 - The title of Table 6 must be placed at the top, an aspect that must be corrected in the remaining tables of the manuscript. 

 Line 151 and 173 - The objectives of the work must appear only at the end of the introduction and not appear duplicated.

 Materials and Methods

Given that this is a review study, the methodology used in the review should be clarified. The type of study and review criteria must be clearly mentioned.

In the organization of the study, point 3 regarding the results needs to be included. There is information in the methodology that can be included in the results.

Discussion 

Authors must include the limitations of the study.

References

While references are adequate, authors should consider including some more recent articles. Of the 31 references, only 12 are under five years old, including a self-citation.

 Good luck with your investigation. 

As long as I can evaluate as a non-native English speaker, the language is adequate but needs improvemen of some grammatical inaccuracies.

Author Response

Dear Reviewer, 
Thank you for your advice. We have edited the manuscript following your advice.  

  • Abstract. Although it is a review article, the abstract should be organized in a structure: Introduction, objective, methodology and conclusions.

We have organized the abstract in a structure: Introduction, objective, methodology and conclusions.

  • Introduction. Line 149 - The title of Table 6 must be placed at the top, an aspect that must be corrected in the remaining tables of the manuscript. Line 151 and 173 - The objectives of the work must appear only at the end of the introduction and not appear duplicated.

We have moved the titles of each table above it. The specific objectives of the work are now stated at the end of the introduction.

  • Materials and Methods. Given that this is a review study, the methodology used in the review should be clarified. The type of study and review criteria must be clearly mentioned. In the organization of the study, point 3 regarding the results needs to be included. There is information in the methodology that can be included in the results.

Our study is not a review study. We have analysed what was already present in the literature for the psychic and gynaecological fields only in order to motivate the choice of the proposed parameters. Precisely because it is not a review study, the results are represented by the table proposed in the "Discussions" section. In fact, the gynaecological field in the "Materials and Methods" reports data, whereas the psychological field reports an analysis methodology. Therefore, we have changed the title of the "Discussions" paragraph to "Results and Discussions".

  • Discussion. Authors must include the limitations of the study.

We have include the limitations of the study. 

  • References. While references are adequate, authors should consider including some more recent articles. Of the 31 references, only 12 are under five years old, including a self-citation.

We have not included more recent articles because for the gynaecological and psychological part, we thought it appropriate to include only the most relevant articles. In the forensic field, on the other hand, there is an absence of important studies such as ours analysing this issue. It is precisely this deficiency that prompted us to propose this study. 

We are very pleased with the proposed revisions and the compliments we have received. We thank you for improving our work, which we hope will lead to an increase in publications in this important area for the forensic community. 

Kind regards, 

Dr. Benedetta Pia De Luca

Reviewer 3 Report

The paper seeks to answer two questions: (1) Is the economic estimate of abortion based on objective and standardized criteria or is it heavily influenced by subjective evaluation? (2) Is it possible to find standard criteria that may act as guidelines to quantify the loss of 30 that human capital ‘in progress’?

The paper is well written and the topic of interest. Addressing a few minor issues would further improve the clarity and applicability of the manuscript. Given the broad readership of the journal an additional few sentences at the start of the manuscript addressing why this reflections/studies are important for other countries (international societies) could be helpful. It is worth noting that the different guidelines introduced in different countries depend not only on medical, economic or legal standards, but also on religious or philosophical considerations. In the Introduction (p. 2, line 83-85), the authors could also explain (at least in a footnote) why they mention only a few countries when they present 'the quantification methods of damages for death' and omit other countries (e.g., China, India, Eastern European countries).

Author Response

Dear Auditor, 
Thank you for your advice. We have amended the text according to your opinion. 

  • The paper is well written and the topic of interest. Addressing a few minor issues would further improve the clarity and applicability of the manuscript. Given the broad readership of the journal an additional few sentences at the start of the manuscript addressing why this reflections/studies are important for other countries (international societies) could be helpful. It is worth noting that the different guidelines introduced in different countries depend not only on medical, economic or legal standards, but also on religious or philosophical considerations.

We have amended the manuscript by adding few sentences at the start of the manuscript addressing why this reflections/studies are important for international societies. 

  • In the Introduction (p. 2, line 83-85), the authors could also explain (at least in a footnote) why they mention only a few countries when they present 'the quantification methods of damages for death' and omit other countries (e.g., China, India, Eastern European countries).

We have explain why we omitted some countries. 

Kind regards, 

Dr. Benedetta Pia De Luca